# An asymmetric centromeric nucleosome

**Yuichi Ichikawa[1,2†], Noriko Saitoh[2], Paul D Kaufman[1]\***

[1]Department of Molecular, Cell, and Cancer Biology, University of Massachusetts Medical School, Worcester, United States; [2]Division of Cancer Biology, The Cancer Institute of JFCR, Tokyo, Japan

**Abstract** Nucleosomes contain two copies of each core histone, held together by a naturally symmetric, homodimeric histone H3-H3 interface. This symmetry has complicated efforts to determine the regulatory potential of this architecture. Through molecular design and in vivo selection, we recently generated obligately heterodimeric H3s, providing a powerful tool for discovery of the degree to which nucleosome symmetry regulates chromosomal functions in living cells (Ichikawa et al., 2017). We now have extended this tool to the centromeric H3 isoform (Cse4/CENP-A) in budding yeast. These studies indicate that a single Cse4 N- or C-terminal extension per pair of Cse4 molecules is sufficient for kinetochore function, and validate previous experiments indicating that an octameric centromeric nucleosome is required for viability in this organism. These data also support the generality of the H3 asymmetric interface for probing general questions in chromatin biology.
DOI: https://doi.org/10.7554/eLife.37911.001

*For correspondence:
paul.kaufman1@umassmed.edu

Present address: †Division of Cancer Biology, The Cancer Institute of Ariake, Tokyo, Japan

Competing interests: The authors declare that no competing interests exist.

## Introduction

The histone octamer is comprised of two copies of each of the four core histones H2A, H2B, H3, and H4, organized with two H2A/H2B dimers associated with a central core tetramer of histone H3 and H4 (*Kornberg and Lorch, 1999*; *Kornberg and Thomas, 1974*). The presence of two copies of each histone in the octamer (*Luger et al., 1997*) raises the potential for three distinct stoichiometries (0, 1 or 2) for any modification, and increasing evidence supports the existence of asymmetrically-modified nucleosomes (*Li and Shogren-Knaak, 2008*; *2009*; *Voigt et al., 2012*). Using rational design, followed by in vivo optimization and selection, we recently developed a pair of H3 proteins – H3X and H3Y – that form obligate heterodimers (*Ichikawa et al., 2017*). Shortly thereafter, another group reported an asymmetric H3 interface (H3D and H3H) based on an electrostatic interaction between a single altered amino acid on each half of the heterodimeric pair (*Zhou et al., 2017*). These systems were used to probe the biological outcomes resulting from packaging of the genome with nucleosomes bearing single or double point mutations, and the findings in the two studies display many similarities. For example, both studies showed that a single H3 N-terminal tail carrying H3K36me3 is sufficient to suppress cryptic transcriptional initiation. Thus, asymmetric nucleosome systems represent a novel approach to mechanistically probe the role of histone stoichiometry in multiple aspects of chromatin biology.

Histone isoforms are an important source of chromatin diversity. Notably, there is a specialized H3 isoform termed CENP-A (or Cse4 in budding yeast) required for centromere function (*Foltz et al., 2006*; *Meluh et al., 1998*; *Stoler et al., 1995*; *Sullivan, 2001*). Here, we extended the utility of our asymmetric H3 interface to explore the stoichiometry of Cse4 domains required for centromeric nucleosome function. The question of Cse4 stoichiometry has received particular attention (reviewed in [*Biggins, 2013*]). Octameric Cse4 nucleosomes, with two copies of Cse4 in place of H3 can be reconstituted from recombinant components in vitro (*Camahort et al., 2009*; *Dechassa et al., 2011*; *Kingston et al., 2011*) and display a canonical nucleosome-like structure when analyzed at high resolution via X-ray crystallography (*Sekulic et al., 2010*). Microscopy-based

experiments (*Aravamudhan et al., 2013*; *Wisniewski et al., 2014*) have also been consistent with octomeric Cse4 nucleosomes. However, crosslinking and atomic force microscopy studies (*Dalal et al., 2007*), reconstitution experiments (*Furuyama et al., 2013*), and chromatin digestion experiments (*Henikoff et al., 2014*) had suggested that Cse4/CENP-A forms a tetrameric half-nucleosome, raising the question of which mode of Cse4 interaction is important for chromosome segregation. Using our tools, we confirm that an octomeric centromeric nucleosome exists and is essential for viability in budding yeast, and show that important residues on both the N- and C-terminal Cse4 tails function when present at one copy per pair.

## Results and discussion

All eukaryotes use a specialized histone H3 variant (CENP-A, or Cse4 in budding yeast) to form centromeric nucleosomes. CENP-A proteins are essential because they provide the fundamental attachment between chromosomes and kinetochore proteins, thereby ensuring equal chromosome segregation at mitosis (*Biggins, 2013*; *Buchwitz et al., 1999*; *Meluh et al., 1998*; *Palmer et al., 1987*; *Stoler et al., 1995*). Cse4/CENP-A proteins have a histone-fold globular domain similar to that of H3, with more divergent N- and C-terminal extensions (*Chen et al., 2000*; *Keith et al., 1999*; *Tachiwana et al., 2011*). To explore the stoichiometry of domains required for Cse4 function, we have generated a heterodimeric 'Cse4X' and 'Cse4Y' pair (*Figure 1A*).

We first attempted to place the X and Y asymmetric mutations we had developed in the canonical H3 protein onto wild-type Cse4. However, these constructs (termed Cse4X' and Y') would not support viability in the absence of a functional, homodimeric Cse4 protein (*Chen et al., 2000*) (*Figure 1—figure supplement 1*). We reasoned that the X and Y alterations may function only in the context of the canonical H3-H3 interface. Because the asymmetric Y alterations are present in both the α2 and α3 helices of the histone fold domain (*Ichikawa et al., 2017*; *Figure 1A*), we tested the X and Y alterations in a chimeric Cse4 in which key residues in these helices were substituted with canonical H3 residues. We used a chimeric Cse4 ('Chimera' in *Figure 1A*; originally termed the *cse4-337* allele (*Keith et al., 1999*) which had previously been shown to support viability (*Keith et al., 1999*); also see *Figure 1B*). Inserting the X and Y asymmetric residues into this Chimera, we generated the Cse4X and Cse4Y proteins (*Figure 1A*). Notably, cells simultaneously expressing Cse4X and Cse4Y were viable, but those expressing a single one of these alone were not (*Figure 1B*). These data support the idea that at least some of Cse4 functions are performed as part of an octomeric centromeric nucleosome, consistent with previous structural, biochemical and cell biological studies (*Aravamudhan et al., 2013*; *Black and Cleveland, 2011*; *Camahort et al., 2009*; *Dechassa et al., 2011*; *Kingston et al., 2011*; *Wisniewski et al., 2014*; *Xiao et al., 2017*; *Zhang et al., 2012*). Further, these data suggested that the centromeric H3 isoform is amenable to analysis via our asymmetric mutations.

We note that the Chimera protein itself conferred a partial temperature-sensitive growth phenotype, in which colony size was markedly reduced at 37°C, but much less so at 34°C (*Figure 1C*). Adding the X and Y alterations to the Chimera protein very modestly increased temperature sensitivity at 37°C, but did not increase the sensitivity of the cells to benomyl, a microtubule destabilizing drug (*Davidse and Flach, 1977*). To test whether an alternative protein design could provide a less perturbed setting for these studies, we compared the X-Y interface to a different pair of asymmetric H3 constructs (*Zhou et al., 2017*) that were described soon after our study was published. That study tested pairs of oppositely charged electrostatic residues at key contact points between nucleosomal H3 molecules, finding one pair of asymmetric H3s that would support viability. This pair is comprised of 'H3$^D$', which contains an A110D replacement, and 'H3$^H$', which contains a L130H replacement. We generated Cse4 derivatives with either the D or H single amino acid substitutions (*Figure 1—figure supplement 2*, Cse4D' and Cse4H'), and we also made D or H substitutions in the context of the Chimera used above (*Figure 1—figure supplement 2*, Cse4D and Cse4H). Notably, we found that after removing homodimeric Cse4, cells expressing either the Cse4H' or the Cse4H protein without any partner protein were viable and grew robustly, indicating significant homodimerization of this protein (*Figure 1—figure supplement 2B*). This led us to test the reported D-H interface in the context of canonical histone H3. We observed that H3$^H$ would also support growth of cells

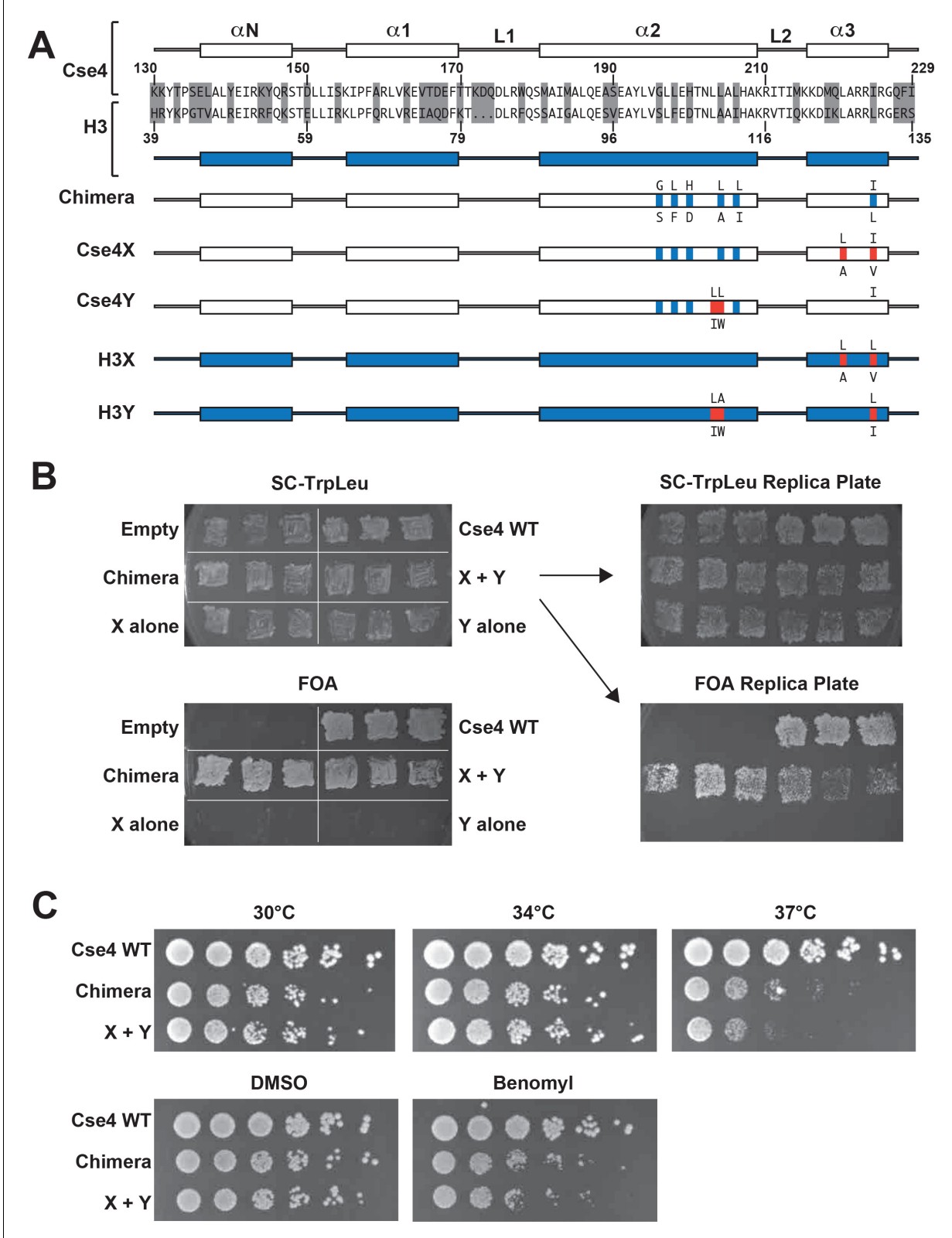

**Figure 1.** Two copies of Cse4 within a nucleosome are required for yeast viability. (**A**) Design of an asymmetric Cse4 interface. Secondary structure and sequence comparison of the Cse4 (white) and H3 (blue) histone-fold domains are shown at the top. Non-identical residues are shaded, and red residues indicate the asymmetric alterations (*Ichikawa et al., 2017*). The Chimera construct includes H3 residues within the context of Cse4 (G196S, L198F, H200D, L204A, L206I, I224L), and this was used to create the asymmetric Cse4X (G196S, L198F, H200D, L204A, L206I, L220A, I224V) and Cse4Y

*Figure 1 continued on next page*

Figure 1 continued

(G196S, L198F, H200D, L203I, L204W, L206I) proteins. The asymmetric H3X (L126A, L130V) and H3Y (L109I, A110W, L130I) proteins are illustrated for comparison. (B) Genetic analysis of heterodimeric Cse4X/Cse4Y pairs. Neither Cse4X alone nor Cse4Y alone support growth. Images show growth of yeast cells upon 5-FOA selection against a *URA3*-containing plasmid carrying *cse4-107* (*Chen et al., 2000*), comparing strains expressing wild-type Cse4, Chimera, Cse4X alone, Cse4Y alone, both Cse4X and Cse4Y, or no Cse4 (empty vector). Colonies were picked from selective media and patched on SC-TrpLeu and FOA plates simultaneously (left panels). Three independent transformants for each strain were grown for 3 days on SC-TrpLeu plates or 7 days on FOA plates. Right panels show replica plating controls to ensure adequate numbers of cells were analyzed. The primary SC-TrpLeu plate (upper left) was replica plated onto SC-TrpLeu as a positive control for cell transfer and onto FOA to test for Cse4 function; both replica plates were incubated for three days. Note that no growth on FOA was observed for any of the Cse4X alone and Cse4Y alone isolates. (C) Growth assay for the indicated strains under stress conditions. (Top row) Serial dilutions of the indicated strains were plated on YPD plates and were incubated at 30°C, 34°C or 37°C for 2 days. (Bottom row) Cells were plated on YPD, YPD +0.2% DMSO or YPD +10 µg/ml benomyl with 0.2% DMSO and were incubated at 30°C for 2 days. Yeast carrying Chimera and Cse4 X + Y nucleosomes grow slower than wild-type (Cse4 WT) at 37°C, and both strains are slightly sensitive to benomyl treatment relative to wild-type. Strains analyzed were: Cse4 WT (PKY5230), Chimera (PKY5232), Cse4 X + Y (PKY5234).

DOI: https://doi.org/10.7554/eLife.37911.002

The following figure supplements are available for figure 1:

**Figure supplement 1.** Simple insertion of asymmetric residues into Cse4 results in non-functional proteins.

DOI: https://doi.org/10.7554/eLife.37911.003

**Figure supplement 2.** Analysis of paired electrostatic residues within the Cse4 dimerization region.

DOI: https://doi.org/10.7554/eLife.37911.004

**Figure supplement 3.** Genetic analysis of H3$^D$/H3$^H$ and H3X/H3Y pairs.

DOI: https://doi.org/10.7554/eLife.37911.005

**Figure supplement 4.** Temperature-sensitive growth assays.

DOI: https://doi.org/10.7554/eLife.37911.006

**Figure supplement 5.** Biochemical analysis of asymmetric Cse4 nucleosome formation in vivo.

DOI: https://doi.org/10.7554/eLife.37911.007

lacking all other sources of histone H3, although growth was faster when both H and D were present (*Figure 1—figure supplement 3*). In contrast, as we had observed before (*Ichikawa et al., 2017*), no growth of cells expressing only H3X or H3Y was observed. Not only do our findings imply significant H3$^H$ homodimerization in vivo, but we find no improvement in temperature sensitivity in the D-H system, as cells expressing H3$^D$ + H3$^H$ were also unable to grow at 37°C (*Figure 1—figure supplement 4*). We therefore performed all subsequent experiments on Cse4 heterodimerization using the Cse4X and Cse4Y pair.

We also tested for heterodimerization of Cse4X and Cse4Y via biochemical experiments. To do this, we expressed three distinct epitope-tagged X or Y proteins simultaneously in the same cell, and used co-immunoprecipitation to analyze dimerization. We had taken a similar approach in the analysis of H3 (*Ichikawa et al., 2017*). In the present case, the poor solubility of the Cse4 nucleosome (*Skene and Henikoff, 2017*) led us to use a different protocol for the preparation of cell extracts. Specifically, we modified existing protocols and generated high-concentration MNase-treated extracts (*Furuyama and Biggins, 2007*; *Ichikawa et al., 2014*; *Nelson and Fangman, 1979*) as the starting point for immunoprecipitation. Upon anti-FLAG epitope immunoprecipitation, we easily detected the expected heterodimeric interactions (i.e. HA-Cse4Y with FLAG-Cse4X in strain PKY5527 cells and V5-Cse4X with FLAG-Cse4Y in PKY5529 cells) but not homodimeric interactions (*Figure 1—figure supplement 5*). Together, our genetic experiments (*Figure 1B*) and the co-immunoprecipitation experiments (*Figure 1—figure supplement 5*) led us to conclude that we had designed a specific heterodimeric Cse4X-Cse4Y nucleosome.

We next used our asymmetric proteins to analyze Cse4 domain structure. The long N-terminal tail of Cse4 contains an Essential N-terminal Domain (END) between residues 28 and 60 (*Chen et al., 2000*; *Keith et al., 1999*, *Figure 2A*), which is a binding site for the Ctf19-Mcm21-Okp1 'COMA' kinetochore protein complex (*Chen et al., 2000*). Complete removal of the END domain via deletion of the N-terminal 70 residues of homodimeric, wild-type Cse4 generated a protein (Cse4 Δ70) that does not support viability (*Figure 2B*), consistent with previous observations (*Chen et al., 2000*; *Keith et al., 1999*). The END domain was also essential in the context of the Cse4 X-Y system, as cells expressing paired Cse4 proteins containing the Δ70 deletion on both X and Y were also inviable. In contrast, cells with one intact END domain on either Cse4X or Cse4Y were viable. To further assess the key biological function of the centromeric nucleosome, we measured centromeric plasmid

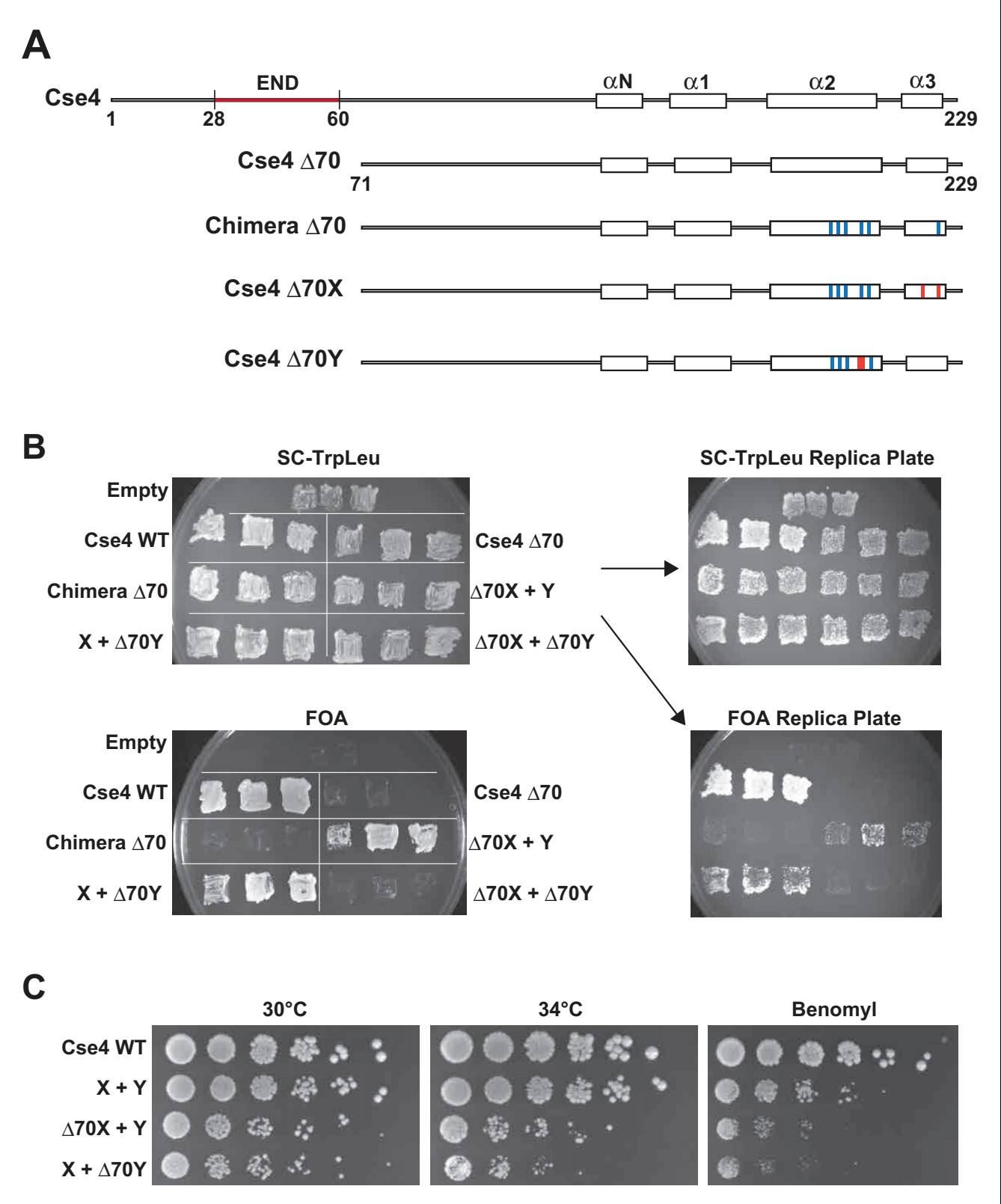

**Figure 2.** A single N-terminal domain within each Cse4 nucleosome is sufficient for yeast viability. (**A**) The location of the Essential N-terminal Domain (END) is diagrammed relative to the alpha helices within the Cse4 histone-fold domain. The N-terminal 70 amino acids containing END were deleted from each construct, as indicated (Δ70 strains). (**B**) Viability test of the Δ70 strains. Deletion of both END regions within a nucleosome was lethal, but a single END per Cse4 nucleosome supports growth. As in *Figure 1B*, images show growth of yeast during selection against a *URA3*-marked plasmid

*Figure 2 continued on next page*

*Figure 2 continued*

carrying *cse4-107*. Three independent transformants expressing the indicated proteins (Cse4 Wild-type, Cse4 Δ70, Chimera, Chimera Δ70, Cse4 X + Cse4 Y, Δ70X + Y, X + Δ70Y or Δ70X + Δ70Y) were grown on 5-FOA plates for 7 days. (**C**) Growth assay for the indicated strains under stress conditions. Serial dilutions of the indicated strains were plated on YPD or YPD +10 µg/ml benomyl with 0.2% DMSO. YPD plates were incubated at 30°C or 34°C for 3 days. The benomyl plate was incubated at 30°C for 3 days. Strains analyzed were: Cse4 WT (PKY5470), X + Y (PKY5476), Δ70X + Y (PKY5485) and X + Δ70Y (PKY5488).

DOI: https://doi.org/10.7554/eLife.37911.008

maintenance, which depends on a cloned centromeric sequence (CEN; [*Murray and Szostak, 1983*]). CEN is the binding site for the Cse4 nucleosome, and is thereby essential for kinetochore-mediated chromosome segregation (*Camahort et al., 2009*; *Meluh et al., 1998*); reviewed in [*Biggins, 2013*]). We observed that cells with a single END-Cse4 construct displayed plasmid loss rates that were statistically indistinguishable from those observed in the presence of the 'pseudo-wildtype' Cse4 X + Y pair (*Figure 3*). These data indicate that a single END domain per Cse4 nucleosome is sufficient for building functional kinetochores. However, these cells displayed partial temperature sensitivity at 34°C and sensitivity to benomyl, suggesting that this architecture is suboptimal under stress conditions (*Figure 2C*). These data confirm that asymmetric Cse4 proteins, like asymmetric H3 proteins, can be used to probe domain stoichiometry in living cells.

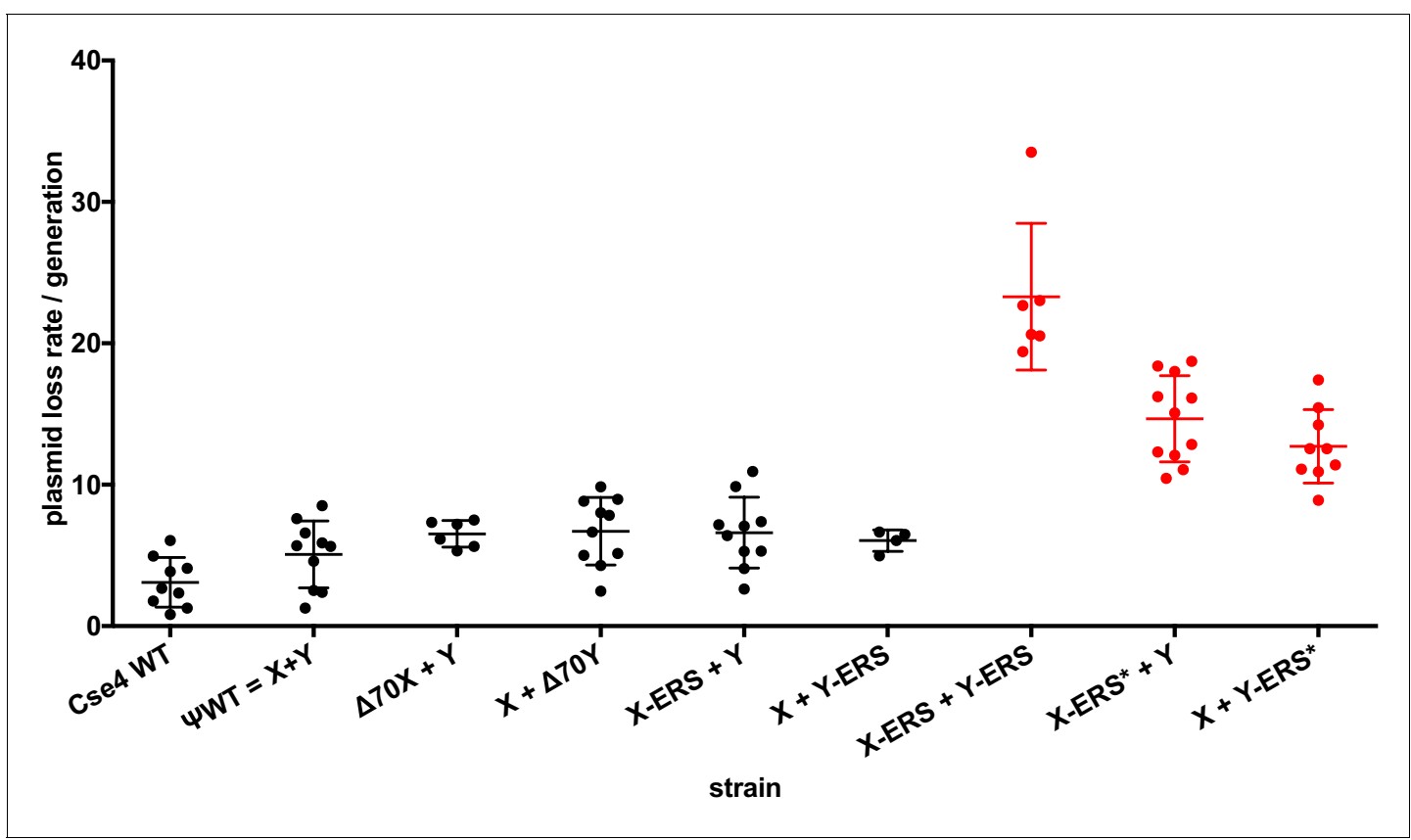

**Figure 3.** Effects of asymmetric Cse4 mutations on plasmid loss rates. Mitotic plasmid stability assay. Test strains bearing plasmid pRS413 (*Sikorski and Hieter, 1989*), which contains a centromere (CEN6), an autonomously replicating sequence (ARS), and the selectable marker *HIS3*, were grown in unselective media (YPD) for approximately 10 generations at 30°C. The fraction of plasmid-containing cells was determined by plating identical aliquots on selective and nonselective media. At least four biological replicates were analyzed for each mutant strain. Individual data points, along with average and standard deviation of the plasmid loss rates are graphed on the y-axis. Constructs that resulted in loss rates significantly different than the pseudo wild-type strain are shown in red (p<0.05, Welch's t test).

DOI: https://doi.org/10.7554/eLife.37911.009

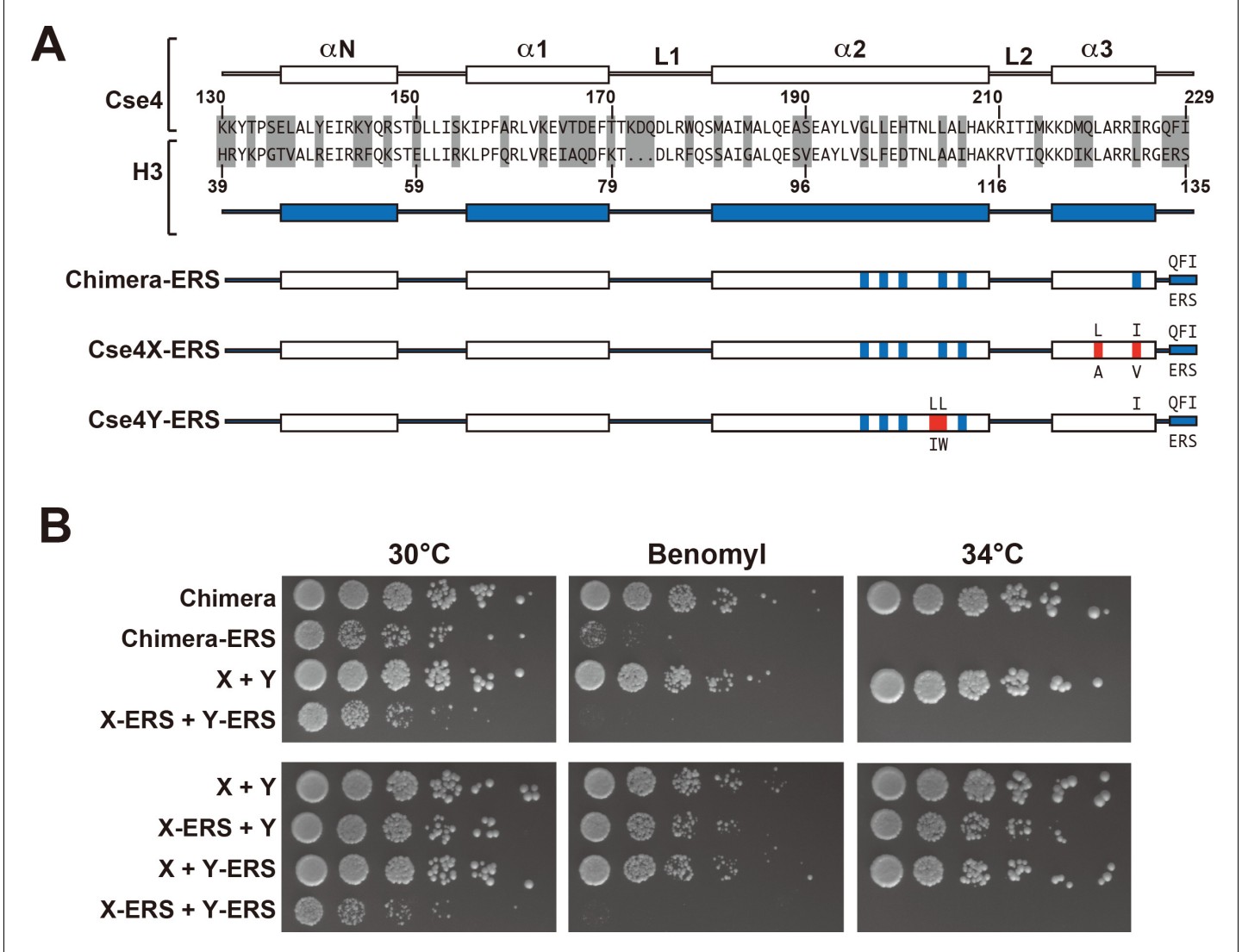

**Figure 4.** A single CENP-C interaction region per Cse4 nucleosome supports viability. (A) As in *Figure 1*, secondary structure and sequence comparison of the Cse4 (white) and H3 (blue) histone-fold domains are shown. ERS mutations (C-terminus QFI/ERS swap) on each construct are indicated. The altered Cse4 or H3 amino acids are shown above the secondary structure, and the substituted amino acids are shown below. Red residues indicate the asymmetric mutations (*Ichikawa et al., 2017*). (B) Growth assay for the indicated strains under stress conditions. (Upper panel) For Cse4 nucleosomes built with the Chimera protein, ERS mutations on both Cse4 molecules resulted in severe growth defects. (Lower panel) In the asymmetric Cse4 nucleosome, a single ERS mutation did not cause significant growth defects, even under stress conditions. Serial dilutions of the indicated strains were plated on YPD or YPD +10 µg/ml benomyl with 0.2% DMSO. YPD plates were incubated at 30°C, 34°C or 37°C for 3 days. The benomyl plate was incubated at 30°C for 3 days. Strains analyzed were: Chimera (PKY5473), Chimera-ERS (PKY5494), X + Y (PKY5476), X-ERS +Y ERS (PKY5503), X-ERS +Y (PKY5497) and X + Y ERS (PKY5500).

DOI: https://doi.org/10.7554/eLife.37911.010

The following figure supplements are available for figure 4:

**Figure supplement 1.** Analysis of additional mutations affecting the CENP-C interaction region.

DOI: https://doi.org/10.7554/eLife.37911.011

**Figure supplement 2.** Combination of N- and C-terminal Cse4 alterations.

DOI: https://doi.org/10.7554/eLife.37911.012

**Figure supplement 3.** Resequencing of plasmids.

DOI: https://doi.org/10.7554/eLife.37911.013

Next, we analyzed motifs at the Cse4 C-terminus. The terminal three residues of Cse4, QFI, are distinct from the ERS residues at the end of H3 (*Figure 4A*), and comprise part of the interaction surface for the centromeric protein CENP-C (Mif2 in budding yeast; (*Kato et al., 2013*), which is critical for kinetochore function (*Meeks-Wagner et al., 1986*; *Przewloka et al., 2011*). We first exchanged ERS residues onto the C-termini of the homodimeric Chimera Cse4 protein and onto the heterodimeric Cse4X and Cse4Y constructs (*Figure 4A*). We observed that cells remained viable in the absence of QFI residues, consistent with previous studies (*Keith et al., 1999*). However, cells expressing only Cse4 proteins with ERS C-termini (e.g. the Chimera-ERS and X-ERS +Y ERS strains) displayed slow growth at 30°C and extreme sensitivity to elevated temperatures or benomyl (*Figure 4B*). X-ERS +Y ERS cells also displayed significantly elevated plasmid loss rates compared to the Cse4 X + Y pseudo-wildtype cells (*Figure 3*). In contrast, X-ERS +Y and X + Y ERS strains with a single ERS domain per Cse4 nucleosome did not display temperature or benomyl sensitivity (*Figure 4B*) or elevated plasmid loss rates (*Figure 3*). Thus, our asymmetric Cse4 tools allowed us to conclude that a single QFI tail per Cse4 nucleosome was sufficient to maintain normal centromere function and prevent stress sensitivity phenotypes.

Recent experiments indicate that in addition to the C-terminus, unique residues within Cse4 Loop1, between the α1 and α2 alpha helices of the histone fold domain, are required for optimal Mif2 binding to Cse4 (*Xiao et al., 2017*). Therefore, we investigated how Loop1 mutations synergize with ERS substitutions in our asymmetric system. Notably, Loop1 in Cse4 is three residues longer than in H3, with an insertion of residues KDQ, and there is also a substitution of an evolutionarily conserved H3 phenylalanine for a tryptophan residue (*Figure 4—figure supplement 1A*). Previous studies have shown that simultaneous deletion of the KDQ residues and conversion of the W to an F residue is tolerated, but results in a ~ five fold increase in chromosome loss rates (*Keith et al., 1999*). In our experiments, combination of the Loop1 and ERS mutations (termed ERS*) in the context of Cse4-Chimera resulted in lethality, as did the presence of the ERS* combination on both halves of the X + Y asymmetric pair (*Figure 4—figure supplement 1B*). In contrast, the combined ERS* mutations could support viability if present only on a single half of the asymmetric Cse4 pair, although there was an increase in plasmid instability, temperature sensitivity and benomyl sensitivity in these strains (*Figure 3*, *Figure 4—figure supplement 1C*). These data are consistent with our earlier conclusion that a single intact Mif2 binding site per centromeric nucleosome is sufficient for viability.

We also tested the effects of combining a single N-terminal deletion (Δ70) and a single set of ERS* substitutions in cis or *trans* configurations within the asymmetric Cse4 nucleosome (*Figure 4—figure supplement 2*). Viable cells in both the *cis* and *trans* mutants could be obtained in the initial selection against the plasmid encoding homodimeric Cse4 (*Figure 4—figure supplement 2A*). We note that the two 'trans' mutant combinations may produce FOA-resistant colonies at different frequencies, perhaps because of combining the X mutations in alpha helix 3 with the C-terminal ERS mutations in the same molecule. In contrast to any small differences between the trans strains, the major observation was that none of the cis or trans strains could be propagated for further analysis (*Figure 4—figure supplement 2B*). These results suggest that simultaneous removal of one interaction site for the COMA complex at the Cse4 N-terminus and one interaction site for CENP-C at the Cse4 C-terminus resulted in a highly compromised Cse4 nucleosome. In other words, a single wildtype N-terminus and C-terminus per Cse4 nucleosome appears insufficient to prevent lethal levels of genome instability.

Additionally, because some of our altered Cse4 proteins were affecting viability and plasmid stability, we wondered if the asymmetric interface was undergoing sufficient negative selection during the course of these experiments to select for revertants different from the input constructs. To test for this, we amplified the *CSE4* genes from the indicated strains listed in *Supplementary file 1* (*Figure 4—figure supplement 3*). In no case did we observe alterations from the input plasmid sequence. We conclude that the phenotypes observed throughout this work result from the designed constructs and not due to intragenic suppressor mutations in *CSE4*.

In summary, we show here that our asymmetric nucleosome technology is well-suited to explore histone H3 variants. In analogy to our H3 studies, these reagents can be used to study Cse4 modifications, for example S33 phosphorylation (*Hoffmann et al., 2018*), K37 methylation (*Samel et al., 2012*) or phosphorylation by Ipl1 (aurora kinase) (*Boeckmann et al., 2013*). Of course, CENP-A stoichiometry and modification is also important in mammalian cells, but involves much more complex

centromeres (*Müller and Almouzni, 2017*), and it will be of great interest to explore nucleosomal symmetry in those systems.

## Materials and methods

### Plasmid constructions and mutagenesis

DNA fragments containing the CSE4/H3 chimera or Cse4X or Cse4Y mutants were synthesized as gBlocks Gene Fragments (Integrated DNA Technologies). To make plasmids containing *cse4* mutants, the DNA fragments and linearized yeast CEN/ARS plasmids (pRS414 or pRS415, [*Sikorski and Hieter, 1989*]) were co-transformed into yeast strain YPH499. N-terminal deletion, L1 and ERS mutations were generated according to inverse PCR based site-directed mutagenesis protocol (Toyobo). DNA sequences of all *cse4* mutant plasmids used in this study were confirmed with Sanger sequencing.

### Yeast strains

All *cse4* mutant strains (listed in *Supplementary file 1*) were derived from *CSE4* 'shuffle' strain PKY2160, which has chromosomal *CSE4*-encoding locus deleted and carries the *cse4-107* gene on a *URA3*-marked plasmid (*Sharp et al., 2002*). PKY2160 was transformed with the test plasmids according to a DMSO-modified version of a yeast LiOAc transformation method (*Hill et al., 1991*). Colonies were picked and streaked on selective media containing 5-FOA to select against the *URA3* plasmid containing the *cse4-107* gene.

### Plate growth assays

For growth assay under high temperature or microtubule destabilization stresses, strains were grown overnight in 2 ml YPD medium at 30°C. Cultures were adjusted to OD600 = 0.6, and five-fold dilutions were spotted on YPD-agar plates or YPD-agar plates containing 10 μg/ml benomyl/0.2% DMSO. Plates were incubated at 30, 34 or 37°C for 2–3 days.

### Mitotic plasmid stability assay

The plasmid loss rate was measured as described previously (*Gibson et al., 1990*) with minor modifications. Briefly, 2 ml YPD cultures were inoculated with fresh colonies picked from selective medium. The cultures were grown for approximately 10 generations. The samples were diluted with water and plated onto selective medium or nonselective medium (YPD). Percentages of plasmid-containing cells were determined from the ratio of colony numbers on selective medium divided by the numbers on nonselective medium. The plasmid loss rate was calculated with following formula:

$N = \log_2(F/I)$

Plasmid loss/generation $= 1-(F/I)^{1/N}$

I is the initial percentage of plasmid-containing cells and F is the final percentage of plasmid-containing cells after N generations in nonselective medium.

### Plasmid resequencing

DNA was purified from yeast strains as described (*Hoffman and Winston, 1987*). DNA was PCR-amplified with oligonucleotide OYI38 (GATATGATTTATCTTGATCCCACTGTGTCG) plus either OYI39 (ATTTAAGTATTGTTTGTGCACTTGCCTATG) to amplify the *CSE4* gene on the *TRP1* plasmid, or plus OYI40 (TGTGGATATACTAGAAGTTCTCCTCGACCG) to target the *CSE4* gene on the *LEU2* plasmid. PCR products were purified on Zymo Research Clean and Concentrate-5 columns and analyzed by Sanger DNA sequencing/capillary electrophoresis, using oligonucleotide OPK1597 (A TGACCTTATAATAACCTTATTTAAAACAT).

### Biochemistry

Immunoprecipitation of Cse4 nucleosomes was performed with yeast cells expressing 3xFLAG-Cse4X, 3xV5-Cse4X and 3xHA-Cse4Y (PKY5527), or 3xFLAG-Cse4Y, 3xV5-Cse4X and 3xHA-Cse4Y (PKY5529). Yeast nuclei preparation was performed as described previously (*Furuyama and Biggins, 2007*). The final pellet including nuclei was resuspended in digestion buffer (10 mM HEPES-NaOH (pH 7.5), 0.5 mM MgCl$_2$, 0.05 mM CaCl$_2$, 1 mM PMSF, 1.8 ml / g of wet cells)(*Ichikawa et al.,*

*2014*). The nuclei were prewarmed at 37°C for 5 min, and then MNase (final 100 units/ml, Worthington Biochemicals) was added at 37°C for 10 min. Reactions were immediately stopped with 11 mM EDTA, and the nuclei were pelleted at 10,000 x *g* for 10 min at 4°C. The supernatant was recovered and mixed with equal volume of high salt buffer (50 mM HEPES-KOH (pH7.5), 500 mM NaCl, 1 mM EDTA, 1% Triton X-100) before immunoprecipitation. 1 ml of the resulting sample was incubated with 10 µl of anti-FLAG M2 affinity gel beads (Sigma-Aldrich) overnight at 4°C, and the beads were recovered and washed three times with the high salt buffer. Proteins were eluted by boiling in 2 x SDS-PAGE buffer (0.1 M Tris-HCl (pH 6.8), 2% SDS, 20% glycerol, 0.02% bromophenol blue, 1/50 vol of 2-mercaptoethanol). For immunoblotting, the eluted proteins were separated by electrophoresis on 10–20% gradient gels (FUJIFILM Wako Pure Chemical Corporation) and transferred onto a nitrocellulose membrane (Amersham Protran Premium 0.2 µm NC (GE Healthcare). The membrane was probed sequentially with the following primary antibodies: anti-HA (Sigma-Aldrich, 12013819001), anti-V5 (Abcam, ab9113) and anti-FLAG (Sigma-Aldrich, A8592). WB Stripping Solution (Nacalai Tesque, Inc.) was used to remove antibodies from the membrane between each probing step. The specific bands were detected with horseradish peroxidase-conjugated secondary antibodies in Chemi-Lumi One Ultra solution (Nacalai Tesque, Inc.), and the images were taken with Amersham Imager 600 (GE Healthcare).

## Acknowledgements

This work was supported by NIH grants R01GM100164 and R35GM127035 (PDK), and JSPS KAKENHI 18H05531 (NS). We thank Oliver Rando for insightful comments; Dr. Jin-Qiu Zhou for providing the histone shuffle strain LHT001; and Dr. Mitsuhiro Shimizu for his great help and precise advice about yeast nuclei preparation.

## Additional information

### Funding

| Funder | Grant reference number | Author |
| --- | --- | --- |
| National Institutes of Health | R01GM100164 | Paul D Kaufman |
| National Institutes of Health | R35GM127035 | Paul D Kaufman |
| Japan Society for the Promotion of Science | JSPS KAKENHI 18H05531 | Noriko Saitoh |

The funders had no role in study design, data collection and interpretation, or the decision to submit the work for publication.

### Author contributions

Yuichi Ichikawa, Conceptualization, Investigation, Methodology, Writing—original draft, Writing—review and editing; Noriko Saitoh, Supervision, Funding acquisition, Writing—review and editing; Paul D Kaufman, Conceptualization, Supervision, Funding acquisition, Investigation, Writing—original draft, Writing—review and editing

### Author ORCIDs

Yuichi Ichikawa https://orcid.org/0000-0002-6773-2455
Paul D Kaufman http://orcid.org/0000-0003-3089-313X

### Decision letter and Author response

Decision letter https://doi.org/10.7554/eLife.37911.018
Author response https://doi.org/10.7554/eLife.37911.019

## Additional files

**Supplementary files**

• Supplementary file 1. Strain list.
DOI: https://doi.org/10.7554/eLife.37911.014

• Transparent reporting form
DOI: https://doi.org/10.7554/eLife.37911.015

### Data availability

All data generated or analysed during this study are included in the manuscript and supporting files.

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
