## [Decision Letter]

Thank you for submitting your article "An asymmetric centromeric nucleosome" for consideration by *eLife*. Your article has been reviewed by three peer reviewers, including Jerry L Workman as the Reviewing Editor and Reviewer #1, and the evaluation has been overseen by Kevin Struhl as the Senior Editor.

The reviewers have discussed the reviews with one another and the Reviewing Editor has drafted this decision to help you prepare a revised submission.

Summary:

Inherent symmetry in histones within nucleosomes prevents clear analysis of redundancy and functional asymmetry in chromatin signaling. Here Ichiwana and Kaufman adapted a strategy for imposing obligate asymmetric H3 nucleosomes in vivo to the yeast centromeric H3 variant, Cse4. To do so, they combined their previously identified asymmetric mutation pair with a chimeric Cse4 in which the key Cse4 residues at the Cse4-Cse4 interface that differ in between Cse4 and H3 were replaced with the canonical H3 sequence (Cse4X and Cse4Y). The authors then used this system to show the following:

1) the centromeric nucleosome is octameric;

2) viability requires one copy of the N-terminal END domain per centromeric nucleosome, though cells are less stress resistant with only one copy;

3) likewise, viability requires one copy of the CENP-C/Mif2 binding site, though again cells are less stress resistant and have a tendency to drop plasmids.

The authors also demonstrate that a competing design for obligate H3-H3 asymmetry is not fully asymmetric in their experiments.

This work represents a new application of the authors' previous technology to study centromeric nucleosomes however is less broadly applicable and comprehensive than their previous studies of canonical H3.

Essential revisions:

1) Biochemical analysis of asymmetric centromeric nucleosomes in vivo could be improved. The authors should perform DMS crosslinking of nucleosomes with differential tags as described previously in characterizing H3X-H3Y orthogonal pair. If this is not possible due to centromeric nucleosome copy number, in vitro characterization of the degree of orthogonality would strengthen the technology.

2) The relevance of the plasmid loss phenotype in Figure 2 is not discussed and should be explained in the context of centromeric nucleosome function.

3) In Figure 4—figure supplement 2, the mutant pairs do not seem to have equivalent severity when swapped (i.e. one trans mutant v. the opposite trans mutant). If this is a reproducible finding, it warrants a comment that swapped pairs need to be examined in future experiments as it suggests that the X and Y proteins do not behave identically to each other.

4) In the Introduction the authors write that they confirm that the centromeric nucleosome is octameric. However, they do not provide sufficient context to justify why this is an issue. Although they cite some work later in Results, it would be helpful to cite some in the Introduction, including papers that suggested that this is a tetramer in *Drosophila*, such as Dalal et al., 2007.

5) Figure 1B – the viability tests lack a positive control. For the Cse4X and Cse4Y tests, we're shown three empty patches with no controls. Here's an example of how to present the results in a more convincing fashion: patch the three transformants alongside another transformant with the X+Y as a positive control and yet another with empty vector as a negative control. Then show replica plating of this one plate to 5FOA and to a plate where all the patches will grow.

6) Similar controls are needed in several other places in this manuscript: Figure 1—figure supplement 1B, Figure 1—figure supplement 2B, Figure 2B, Figure 4—figure supplement 1B, and Figure 4—figure supplement 2A.

7) Were all yeast transformants sequenced to be sure they had the correct plasmids and that no recombination had taken place between the two copies of cse4?

---

## [Author Response]

Essential revisions:1) Biochemical analysis of asymmetric centromeric nucleosomes in vivo could be improved. The authors should perform DMS crosslinking of nucleosomes with differential tags as described previously in characterizing H3X-H3Y orthogonal pair. If this is not possible due to centromeric nucleosome copy number, in vitro characterization of the degree of orthogonality would strengthen the technology.

Indeed, we tried to do DMS crosslinking, but we were not successful in observing crosslinked species for Cse4. We don’t know if this is because of the copy number, or because of the poor solubility of Cse4 nucleosomes (Skene and Henikoff, 2017). However, when we modified some existing protocols and generated high-concentration MNase-treated extracts (Nelson and Fangman 1979, Furuyama and Biggins 2007, Ichiakawa et al., 2014) as the starting point for immunoprecipitation experiments, we captured asymmetric Cse4 nucleosomes, whose composition was consistent with hetero- but not homodimerization (now Figure 1—figure supplement 5). These data, together with our genetic experiments, make us confident that we have generated heterodimers in vivo.

2) The relevance of the plasmid loss phenotype in Figure 2 is not discussed and should be explained in the context of centromeric nucleosome function.

The plasmid loss data is in Figure 3. We have now explained the importance of the centromeric nucleosome to chromosome maintenance more thoroughly in the fifth paragraph of the Results and Discussion.

3) In Figure 4—figure supplement 2, the mutant pairs do not seem to have equivalent severity when swapped (i.e. one trans mutant v. the opposite trans mutant). If this is a reproducible finding, it warrants a comment that swapped pairs need to be examined in future experiments as it suggests that the X and Y proteins do not behave identically to each other.

Our data doesn’t support the contention that the X and Y halves of the asymmetric Cse4 nucleosome are intrinsically different. Specifically, in our most quantitative analyses of X and Y (e.g. the plasmid loss experiments in Figure 3), we detect no evidence for functional differences between the two possible X and Y configurations.

However, we agree that the two “trans” mutant combinations in Figure 4—figure supplement 2A may produce FOA resistant colonies at different frequencies. One possibility to explain this would be if combining the X mutations in α helix 3 with the C-terminal ERS mutations in the same molecule caused a synergistic weakening of the protein. However, we feel that further attempts to measure what are probably small differences between the two different trans configurations is beyond the scope of this paper. We have now emphasized in the text that the major result here is in Figure 4—figure supplement 2B, in which both cis and trans configurations fail to generate strains that can be propagated. In other words, a single wild-type N-terminus and C-terminus per Cse4 nucleosome is insufficient to prevent massive genome instability.

4) In the Introduction the authors write that they confirm that the centromeric nucleosome is octameric. However, they do not provide sufficient context to justify why this is an issue. Although they cite some work later in Results, it would be helpful to cite some in the Introduction, including papers that suggested that this is a tetramer in Drosophila, such as Dalal et al., 2007.

We have now discussed the previous literature on this subject in more detail in the last paragraph of the Introduction.

5) Figure 1B – the viability tests lack a positive control. For the Cse4X and Cse4Y tests, we're shown three empty patches with no controls. Here's an example of how to present the results in a more convincing fashion: patch the three transformants alongside another transformant with the X+Y as a positive control and yet another with empty vector as a negative control. Then show replica plating of this one plate to 5FOA and to a plate where all the patches will grow.

Responding to this and the following comment, we now include this type of analysis throughout the paper. None of the previous conclusions have changed.

6) Similar controls are needed in several other places in this manuscript: Figure 1—figure supplement 1B, Figure 1—figure supplement 2B, Figure 2B, Figure 4—figure supplement 1B, and Figure 4—figure supplement 2A.

See previous comment.

7) Were all yeast transformants sequenced to be sure they had the correct plasmids and that no recombination had taken place between the two copies of cse4?

We have now recovered DNA from the strains listed in Supplementary Table 1, and sequenced the CSE4 genes amplified using primers that separately anneal to either the TRP1 or the LEU2 plasmid in each strain. These results are presented in Figure 4—figure supplement 3. In all cases, the sequence of the input plasmids was maintained exactly.